# Mediating reconciliation with God: Exploring divine forgiveness experiences during confession among Catholic priests from four Spanish-speaking countries

Carmen Callizo [1,2], Martiño Rodríguez-González [1]*, María Calatrava[1],
María Pilar Martínez-Díaz [3], Kaye Cook[4], Richard G. Cowden[2,5]

1 Institute for Culture and Society, University of Navarra, Pamplona, Spain, 2 Human Flourishing Program, Harvard University, Cambridge, Massachusetts, United States of America, 3 Faculty of Humanities and Social Sciences, Department of Psychology, Comillas Pontifical University, Madrid, Spain, 4 School of Psychology and Human Services, Gordon College, Wenham, Massachusetts, United States of America, 5 Department of Epidemiology, Harvard T.H. Chan School of Public Health, Boston, Massachusetts, United States of America

* mrgonzalez@unav.es

## Abstract

Divine forgiveness (DF), the subjective experience of being forgiven by a higher power after a transgression, remains understudied in psychology, particularly across diverse cultural and religious contexts. This study examines priests' experiences and understandings of DF, both as confessors and penitents, in the Catholic Sacrament of Confession through semi-structured interviews with ten Spanish-speaking priests from Spain, Mexico, Peru, and Chile. Guided by a constructivist stance and a Christian-sensitive relational spirituality model of personal sin, we conducted a theory-informed framework analysis combining inductive coding with iterative matrix-based synthesis. Findings showed that 1) confession emerged as a relational and dynamic encounter involving the self, sin, God, others, and the confessor, whose mediating role makes DF tangible; 2) DF unfolded across phases of vulnerability, repentance, confession, penance, and absolution; 3) barriers included guilt, shame, and scrupulosity, whereas facilitators involved empathetic priestly presence and communal support; and 4) participants distinguished between cognitive certainty of DF and its emotional realization enabled by the confessor's presence. These findings provide preliminary, contextually grounded insights related to both Christian-sensitive and faith-neutral models of DF, which should be further examined and empirically tested in future research.

## Introduction

The experience of divine forgiveness (DF)—understood as the subjective sense of being absolved by a supreme being or higher power following transgression—is

**Data availability statement:** The dataset is available at OSF via https://doi.org/10.17605/OSF.IO/9V5BW. The repository includes the fully anonymized interview transcripts analyzed in this study, the interview guide used as study material, and the coding matrix developed during the analysis from which the reported themes were derived. In accordance with ethical approval from the Ethics Committee of Universidad Pontificia Comillas (approval no. 39/23-24) and applicable data protection regulations (Ley Orgánica 3/2018; GDPR 2016/679), all materials have been fully anonymized to safeguard participant confidentiality and respect the sacramental seal of confession.

**Funding:** The authors disclosed receipt of the following financial support for the research, authorship, and/or publication of this article: This research was supported by the John Templeton Foundation under Grant 61927 awarded to Frank Fincham and Myriam Rudaz; subaward to Martiño Rodriguez-González and María Calatrava.

**Competing interests:** The authors have declared that no competing interests exist.

central to many religious traditions and may influence well-being [1–4]. While forgiveness research has advanced considerably in the domains of interpersonal and self-forgiveness [5], empirical studies on the psychology of DF remain scarce, with limited evidence across methods and populations [1]. This gap hinders understanding of a potentially widespread experience, given that most of the global population believes in God or a higher power [6]. To examine DF psychologically, an in-depth inquiry is required to explore how it is experienced in specific contexts. This study contributes to addressing this gap through a qualitative analysis of DF and reconciliation with God via Catholic confession, drawing on interviews with Spanish-speaking priests and engaging contemporary models of DF [3,7].

## Psychological models of the search for and experience of DF

Two main psychological models have been proposed to explain how people seek and experience DF following the recognition of a personal transgression. Firstly, Fincham and May [3] propose a heuristic model for seeking DF, designed for use across faith traditions but modulated by each tradition's specific emphases. This model describes a series of decisions shaped by cognitive, emotional, and cultural factors. It begins with the perception of a transgression, which elicits emotions such as guilt or shame and motivates the pursuit of DF. The first decision point refers to whether or not the person chooses to seek DF [8]. This decision may be influenced by factors such as the image of God, spiritual attachment, perceptions of divine justice, self-control, and cultural norms [1,9,10]. If they proceed, the second decision concerns whether DF is perceived as unconditional or contingent upon factors such as rituals like confession or repentance [11]. Upon making these decisions, individuals may experience DF, often accompanied by relief, peace, shifts in self-perception, and changes in the relationship with God. The model is dynamic, potentially involving repeated cycles of seeking and context-dependent variations, with outcomes subject to individual interpretation [3].

Secondly, Cook and Cowden [7] recently approached DF within Christianity as a relational experience of reconciling with God following instances in which the individual has become aware of personal sin. Drawing on the framework of relational spirituality [12,13], their model posits that believers can cultivate an active, meaningful relationship with the sacred. DF is conceptualized as arising from a dynamic system in which intrapersonal (e.g., beliefs), interpersonal (e.g., relationships with others), and transpersonal (e.g., perceptions of God) factors interact to support relational homeostasis with God. In the context of religious transgression, individuals engage in cognitive-emotional appraisals involving the self, God, personal sin, and others implicated, as well as the connections among them. These elements are represented in a pyramid structure (see Fig 1), which depicts the relational dynamics of seeking and experiencing reconciliation with God after a personal sin. Based on interdisciplinary insights from theology and philosophy, Cook and colleagues also introduce the notion of experiencing reconciliation with God as involving two interrelated dimensions [14,15]: engaging in repentance (internal acknowledgment of wrongdoing and openness to transformation) and experiencing absolution (a sense of receiving

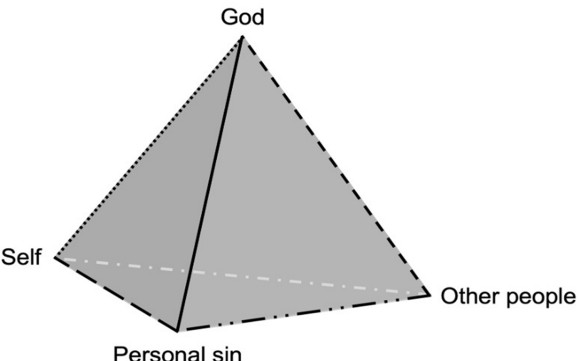

**Fig 1. Relational spirituality model of a Christian's reconciliation with God after recognizing personal sin.** The figure illustrates Cook and Cowden's [7] Christian-sensitive relational spirituality model of personal sin. The vertices represent the self (the individual who has transgressed and seeks divine reconciliation), God (the transcendent figure who offers forgiveness and whose image shapes the experience), personal sin (the specific act or attitude perceived as a rupture in the relationship), and others (such as the religious community or those affected by the sin). The edges depict appraisals of the relationships among these elements following a transgression, highlighting the systemic and relational nature of DF.

forgiveness from God after not fully experiencing it), both in dialogue with the individual's perceptions of God, sin, and others. The model emphasizes that awareness of personal sin may be experienced as a form of rupture to the person's relationship with God, but also marks the start of a relational, contextually embedded process of restoration.

Although the two previous DF models are beginning to receive empirical support, their applicability across different cultural and denominational settings remains largely unexplored. Fincham and Maranges [8] provided initial evidence for the Fincham and May [3] model with predominantly White, English-speaking U.S. Christian university students, while Cook, Cowden, and colleagues [7,14] developed their theologically-informed model through an initial qualitative phase with Christians from 10 different countries, followed by a pilot study with U.S. Christians, all in English. The authors of both models explicitly encourage future research to examine these frameworks in diverse cultural and religious contexts. Within Christianity, Cook and Cowden [7] note Catholicism as a specific case due to its emphasis on sacramental confession in receiving DF.

### The experience of DF in Catholicism

DF holds a central place in Christianity [14], and within it, Catholicism emphasizes a distinct and significant mode of experiencing it through the Sacrament of Penance and Reconciliation [16]. This sacrament, required at least once a year for Catholics, involves confessing sins to a priest as a condition for receiving God's forgiveness [17], thereby making the relational nature of forgiveness explicitly manifest through the presence of an intermediary and situating reconciliation within a broader communal horizon [18]. It aims to restore the relationship between the believer and God, allowing the penitent (i.e., the person who wants to confess a sin) to experience acceptance and reconciliation after failure [19], and it may also influence well-being [20–23]. The rite follows a structured sequence: conversion (awareness and repentance), confession (verbal admission to a priest), penance (acceptance of a reparative act or intention), and absolution (receiving DF). The experience is mediated by the confessor, who represents both the religious community as a human and, during absolution, acts in persona Christi—in the person of Christ—thus acquiring a divine role [16,17,24,25].

Concerning existing models of seeking and experiencing DF, Catholic confession could present distinctive features. In Fincham and May's [3] model, sacramental confession could be a clear example of conditional DF, involving both the decision to seek absolution and the belief that rituals—such as confessing to a priest and performing penance—are required. In the relational spirituality model of personal sin [7], the sequential stages of confession (conversion, verbal admission,

penance, and absolution) could be seen as corresponding to the movements of repentance and absolution that drive the experience of reconciliation [14]. In addition, confession could bring together all the key elements of the model and their relationships: the self (penitent), the confessed sin, the others (the community and those affected by the sin), and God. Thus, an open question is whether the confessor is perceived to belong within the pyramid model shown in Fig 1, as he could be placed in several positions—for example, as a member of the community (the "others" vertex), as a representative of God ("God"), or as a fourth foundational vertex, creating a quadrangular pyramid that links and integrates all dimensions of the sacramental experience. However, these possibilities remain unexplored.

Despite the centrality of confession in Catholicism, qualitative research on the psychological experience of DF in this sacrament is scarce. Some studies explored confession or disclosure to spiritual leaders' experiences in other Christian contexts [26], but these practices differ from the Catholic context, where confession constitutes a formal sacrament, is doctrinally required, and is institutionally embedded as a core element of religious life [17]. One exception that specifically examined experiences of Catholic confession is Wise's [27] doctoral dissertation, which identified themes such as motivation, guilt, shame, scrupulosity, and absolution in experiences of DF, although it focused solely on the perspective of penitents, not confessors. Moreover, those studies have focused on English-speaking contexts. However, findings from these settings should not be uncritically generalized to predominantly Catholic, Spanish-speaking contexts, where cultural factors like family centrality, communal values, and relational orientation are deeply rooted and may shape the way religion is lived, including the experience of confession and DF [28–30].

Examining Spanish-speaking priests' understandings and experiences as both confessor and penitent (confessing their sins with other confessors) from their own perspective could offer a unique lens [24], helping to clarify the confessor's perceived relational significance [7] and the conditional nature of seeking and receiving DF [3]. To our knowledge, no prior studies have examined these dynamics in Spanish-speaking, predominantly Catholic contexts from the confessor's own perspective. The lack of research from this perspective likely reflects both the relative infancy of empirical research on DF and the sensitivity of the topic due to the sacramental seal, which requires the confessor to keep information about the content of the confession and identity of the penitent completely confidential [17]. Nonetheless, research that respects confidentiality may illuminate crucial gaps.

This study focuses on experienced priests who are native to Spanish-speaking contexts, a population underrepresented in psychological research [31] but especially relevant in the present study for two reasons: the predominance of Catholicism in these countries [32] and their communal cultural values, which give religious experience a particularly relational and interdependent character [29,30]. Our work seeks to contribute a more culturally grounded understanding of the experience of reconciliation with God in Catholic contexts, expanding the emerging field of research on DF and informing the development of theoretical models that are sensitive to the particularities of specific faith traditions.

## Objective

This study aimed to explore how Spanish-speaking Catholic priests understand and experience DF within the Sacrament of Confession, considering their perspectives both as confessors (when administering the sacrament) and as penitents (when confessing to other priests). The main goal was to illuminate how the elements of the relational spirituality model of personal sin (i.e., the self, God, the others, and personal sin) [7] are both manifested and interrelated within the experience of confession, and to examine the particular role of the confessor among these elements. It further aimed to broaden understanding of how priests experience and conceptualize DF, considering the relational, cognitive, and emotional factors they identify as facilitating, hindering, or conditioning it—thereby contributing insights that may also complement the DF model proposed by Fincham and May [3] within the specific context of Spanish-speaking Catholicism.

## Materials and methods

This study adheres to the consolidated criteria for reporting qualitative research 32-item guidelines [33]. All methodological details corresponding to these criteria are reported throughout the Methods section; a complete summary is provided in the Supporting Information (S1 Table).

### Inclusivity in global research

Additional information regarding the ethical, cultural, and scientific considerations specific to inclusivity in global research is included in the Supporting Information (S1 Checklist).

### Epistemological approach and methodological design

This study employed theory-informed framework analysis design within a constructivist framework, focusing on how priests experience and make meaning of a spiritual phenomenon from a psychological perspective within specific cultural and relational contexts. Instead of addressing DF as a theological doctrine, it explores how Catholic priests experience their role as mediators of forgiveness within the sacrament of confession, as well as how they live out DF when they are penitents with fellow priests.

Framework analysis [34] was particularly appropriate because our study was guided by the Christian-sensitive relational spirituality model of personal sin proposed by Cook and Cowden [7], while remaining open to inductive development. In practice, the analysis was theory-informed, similar in spirit to a theory-driven qualitative content analysis [35], but it retained the structured, matrix-based, and interpretive logic of framework analysis. The analytic framework served as a flexible reference point, allowing new categories to emerge inductively and accommodating denominational and experiential particularities that might not align neatly with the guiding model [7].

### Participants

Participants were ten Catholic priests, including both diocesan priests (i.e., serving in parishes under the authority of a bishop) and members of religious communities (Jesuits, Franciscans, and Opus Dei), which differ in their spiritual traditions and community life. Eligibility criteria required participants to be native Spanish speakers, to be currently serving as priests and confessors in a Spanish-speaking country, and to have at least 15 years of experience since ordination. Participants were recruited through purposive and snowball sampling. All communication was conducted in Spanish, using email, telephone, and WhatsApp. Initial contacts were established through personal networks, followed by referrals from participating priests and outreach to diocesan offices in various cities. As a result, some participants were previously known to the researchers, while others were not. Recruitment was guided by previously defined diversity criteria to ensure a range of perspectives relevant to the aims of the study.

Participants were based in different cities across Spain ($n = 7$), Peru ($n = 1$), Chile ($n = 1$), and Mexico ($n = 1$); their country of origin matched their current place of ministry. Ages ranged from 46 to 66 years ($M = 55$), and time since ordination ranged from 16 to 39 years ($M = 25$). On average, they reported hearing about 25 confessions per week, though this number varied considerably across participants.

### Data collection

Semi-structured interviews were conducted to explore priests' experiences and understandings of DF both as confessors (when administering the sacrament) and as penitents (when confessing to other priests) as well as their perspectives on how other penitents experience confession. The interview guide was developed with input from two Spanish-speaking Catholic priests who were not subsequently interviewed as participants. In addition, two pilot interviews (not included in the analysis) were conducted with other priests to gather feedback on the semi-structured interview and refine the guide before the final round of interviews.

The interviews began with an open-ended question ("Could you tell me about your experience or the process of DF in confession, or anything you would like to share about this topic?"), allowing participants to share their experiences freely, followed by more specific semi-structured questions that built on these accounts to explore how they experience DF as both confessors and penitents, the actions they carried out during confession, their relationships with penitents and other confessors, and their reflections on different types of sins as well as on well-being (the full list of interview questions is available on the Open Science Framework (OSF) website; see Data availability statement).

Interviews were held either in person at the university or online via video call from the priests' parish or community residence, depending on their location. Each lasted between 60 and 99 minutes (*M* = 74 minutes). All interviews were conducted in Spanish between April 2024 and April 2025. Initial contact was made via email, which included an information sheet and a consent form. Interviews were scheduled only after receiving signed consent and began with recorded verbal consent. All interviews were audio-recorded with permission, transcribed verbatim, and anonymized. Participants chose the following unrelated pseudonyms to ensure confidentiality: 17 Junio, Cercanía, Corazón, Felicidad, Independencia, Juan Pablo II, Lic, Piedra, Sacristán, and Santa Rosa. The fully anonymized interview transcripts in Spanish are available (see Data availability statement).

## Data analysis and credibility strategies

Data were analyzed using the framework analysis approach [34,36]. The analytic unit was each meaningful segment of text at the sentence or short-paragraph level, which was identified and segmented within each transcript. First, two researchers (CC and MC) independently familiarized themselves with the data and conducted initial inductive open coding using Taguette (https://www.taguette.org/). After three interviews, an initial analytic framework was collaboratively developed and iteratively refined as new data emerged. The coded data were then indexed within the framework, charted across cases, and clustered into higher-order categories, which were discussed, named, and organized into thematic matrices by category and participant. Finally, we created a table including the various codes (Level 1: specific; Level 2: broader; Level 3: overarching categories) organized into broader narrative themes (see Supporting Information S2 Table).

The analysis was guided by Cook and Cowden's [7] Christian-sensitive model that conceptualizes personal sin within a relational spirituality framework, while allowing new categories to emerge inductively beyond this structure. Thematic saturation—the point at which no new themes or relevant information emerge—was reached after the eighth interview, and analysis of the final two interviews confirmed that no new themes were identified, consistent with qualitative research standards [37].

To support reflexive rigor, ChatGPT-4, a large language model developed by OpenAI [38] was used as an auxiliary analytic aid exclusively for robustness checks and not for primary coding or interpretation. The model did not generate primary codes or analytic categories, nor did it contribute to the development of the interpretive framework. Instead, it was prompted with summaries of our three-level codes and categories, and asked to propose potential reorganizations or refinements. A typical prompt was: "You are assisting with qualitative data analysis. The following are three-level codes derived from interview data, which have been organized into broader thematic categories. Please evaluate this organization indicating if any codes should be moved, merged, or redefined, and briefly explain your reasoning." The outputs were systematically compared with our own interpretations, and suggestions were incorporated only when consistent with the data corpus and theoretical framework. This step resulted in minor adjustments; for example, the model suggested that the 'confession frequency' code might belong to more than one thematic domain, prompting us to discuss whether it fit better within dispositional preparation or the dynamic process of reconciliation. All analytic decisions remained under the full control of the research team and no AI-generated content was included verbatim in the final results.

To ensure analytic credibility and ethical integrity, we used several credibility strategies. Coding was independently conducted by two researchers (MC, CC) and compared with a third (MRG) to resolve discrepancies and reach consensus. Participant validation occurred through in-interview member checking, in which participants were asked to confirm that our

interpretations reflected their intended meanings. At the end of each interview, participants were also invited to clarify, retract, or add any statement—either immediately or within subsequent weeks. All confirmed that their contributions were accurately represented. Theoretical triangulation was also pursued by discussing our emerging themes with KC and RC, the authors of the relational spirituality model of personal sin [7], to assess their coherence with the model's theoretical framework.

### Reflexivity and positionality

The research team was diverse, comprising researchers primarily based at Catholic universities in Spain (CC, MC, MRG, and MPMD), several of whom lead Ibero-American psychology projects and maintain collaborations across Latin America. This subgroup, which conducted the data collection and primary analysis, was immersed in a Spanish-speaking Catholic context that facilitated access to participants and enabled a nuanced understanding of the lived meanings of DF. Because all participants were male priests, interviews were conducted by male researchers (MRG and an external collaborator) to enhance participant comfort and openness. While this shared linguistic, cultural, and institutional background supported rapport and interpretive sensitivity, we recognized that it could also influence theoretical framing and data interpretation. To balance this, we engaged in continuous reflexive dialogue and included two coauthors (KC and RGC) from the United States: One from a Christian but not specifically Catholic institution and another from a secular university. They are also the original authors of the model used in the present study. Their involvement provided deeper conceptual coherence and theoretical warmth. The first author (CC) completed a research stay with the last author (RGC) during the development of this study, which fostered ongoing theoretical dialogue and cross-cultural reflection throughout the analytic process. This intercultural collaboration enriched the analysis and situated the findings within the wider field of DF research. No conflicts of interest or competing interests were reported.

### Ethical considerations

The study was approved by the Research Ethics Committee of Comillas Pontifical University (approval no. 39/23–24). Informed consent was obtained from all participants prior to their participation, both in written and verbal form. Participants were informed of their right to decline to answer any question or to withdraw from the study at any time without consequence. Special care was taken to ensure confidentiality and respect the seal of confession. Data were handled in compliance with Spanish law (Ley Orgánica 3/2018) and the EU General Data Protection Regulation (2016/679).

### Findings

The findings are organized around themes related to the experience of DF in confession, reflecting the perspectives of priests both as confessors and as penitents. Selected interview excerpts are included to illustrate the richness and complexity of the confessional process. All quotations were translated from Spanish to English by a bilingual member of the research team, and subsequently reviewed by another bilingual team member to ensure accuracy and fidelity to the original narratives. The analysis yielded five overarching themes that structure priests' understanding and mediation of DF: 1) Living confession: The relational fabric of experience; 2) Dynamic movements of reconciliation; 3) Perceived barriers and facilitators; 4) Cognition and emotion in the experience of DF; and 5) Integrative axis: The relational fabric and the dynamism of DF. Each theme is presented below together with its corresponding sub-themes. A full synthesis of all themes and sub-themes is provided in Supporting Information (S2 Table).

### Theme 1. Living confession: The relational fabric of experience

According to the inductive analysis, and in subsequent comparison with the relational spirituality model of personal sin [7], confession within the Catholic tradition was described by participants as a complex relational process in which DF is experienced as being integrated through the dynamic interaction between the self, sin, God, and others—supporting the core

elements of the model—but also suggesting a potential extension by introducing the confessor, who could be experienced as occupying an irreplaceable role. The main categories of this thematic axis are presented below.

**The vulnerable self in confession.** Confession begins with the personal vulnerability of the penitent, with the self-recognizing its own fragility and need for acceptance. As one priest shared, the doorway to authentic reconciliation with God is being able to "*be myself, to say I don't have to hide or disguise my wrongdoing*" (Felicidad). This sincerity is experienced as "*the beginning of a process of personal transformation*" (Cercanía). Preparation for confession requires an inner openness to confession and certain prior acts: "*First, it is a grace; the better prepared you are through the acts of the penitent, the more disposed you will be*" (Sacristán). The experience of DF is thus understood as interactive—participants portrayed God's grace as moving the penitent to prepare and confess, while confession itself was described as opening the way for grace. In sum, confession was experienced as an act of honesty in which the vulnerable self opens itself up to God's grace and the possibility of inner change.

**The personal sin as wound and catalyst for seeking.** Personal sin was experienced not only as a fault, but as an existential wound that prompts the search for reconciliation with God. Priests distinguish between different types of personal sin, noting that some are harder to psychologically overcome than others: "*There are generally two types of sins—those of weakness and those of malice. When someone has experienced a certain malice in their sin, sometimes it can be more difficult*" (17 Junio). However, regardless of the type of personal sin, sorrow is seen as a necessary condition for spiritual change: "*If there is no sorrow for sin, it is difficult to have an experience of conversion, of encountering God*" (Corazón). Importantly, this pain is not lived solely as a burden; it also becomes a catalyst for self-knowledge and openness to grace: "*My wrongdoing was also connected to a kind of thirst for happiness [...] it helps you live it with more truth, more light*" (Felicidad). Thus, confession was described as helping to reframe the wound of personal sin by transforming it into an opportunity for self-discovery, growth, and a renewed encounter with God.

**God's presence and grace.** At the core of the confessional experience participants described the encounter with God, which they perceived as grace and acceptance. For the penitent, God's presence, mediated through the sacrament, becomes a source of certainty and relief: "*Christ is the one who listens to you, the one who forgives you [...] this perception helps a lot*" (Felicidad). This assurance of mercy, rooted in the Gospel and said by the priest, could sustain the penitent: "*You speak to them about God's forgiveness, you remind them, for example, of Gospel passages where you see how Jesus forgives… Therefore, if Jesus always forgives, you are not going to be an exception*" (Sacristán). Experiencing God's grace in confession was described as essential for spiritual renewal: "*Confession has been a very important moment of discovering the experience of God [...] The embrace of the Father, who through Jesus lifts you up, restores that feeling*" (Corazón).

**Other people and supportive bonds.** The relational dimension of confession extends beyond the individual's relationship with God and enters the social realm, involving both the people affected by the sin and the religious community, which serves as a reference for experiencing DF. On the one hand, sin that impacts another person often carries a heavier burden of guilt and shame, making it more difficult for penitents to feel forgiven by God. As one priest observed, "*I think perhaps the most difficult sins to forgive, to feel forgiven for, and to forgive oneself for, are those involving harm to others, especially when the harm was done knowingly*" (Felicidad). Thus, the feeling of reconciliation in some cases was described as extending beyond absolution from God to also include seeking forgiveness from the person who was hurt.

On the other hand, social and community ties seem to serve as sources of support and motivation. Many priests emphasized the importance of a vibrant church community: "*People come to confession if they have a strong experience of Church. By contrast, if the Church is withdrawn or in decline, people go to confession less*" (Corazón). Liturgical seasons, such as Lent or Advent, also encourage greater openness to confession. Social relationships can play a decisive role as well: "*Friends bring others to confession. Girlfriends and boyfriends bring others to confession. The contagion of Christian life... influences people to approach the sacrament*" (Corazón). Previous experiences and the community climate can significantly shape one's disposition to seek confession.

Interestingly, the priest is often seen as a representative of the wider religious community and often plays a key role in facilitating experiences of reconciliation with God. This is especially clear in the act of spiritual direction, which differs from confession in both scope and structure. While confession focuses on the forgiveness of sins and is bound by the sacramental seal, spiritual direction offers broader, ongoing accompaniment. It covers all aspects of Christian life: growing in virtue, discerning life choices, and finding encouragement in what is going well. "*True spiritual direction is not just about giving advice, but about walking together, knowing deeply the person's life and struggles*" (Santa Rosa). Its goal is to help penitents live their faith authentically beyond the confessional: "*The aim of spiritual direction is to lead the penitent to total authenticity in faith, so that they live not by appearances but in true connection with God*" (Independencia). For many, it is a privileged space for discernment: "*In my experience, spiritual direction has been crucial for many deep transformations; it's the space where you can really see the work of the Spirit in the soul*" (Lic).

Although sacramental confession and spiritual direction are distinct, in practice they are often closely connected: "*It is common for the confessor to also be the spiritual director, their support*" (Corazón). Some priests believe that when the same person is both spiritual director and confessor, it can enhance the penitent's experience of reconciliation: "*Spiritual direction integrated with confession is like having two wings; one doesn't work well without the other*" (Piedra). Although priests know that the sacramental seal implies that both spaces are different and the confession is a totally independent space, this dual role implies the priest acts as both another member of the community and a unique figure who forms an intimate bond with the penitent in the search for DF.

**The confessor as face of God and humanity.** In the context of confession, the priest assumes a distinct role from that of a spiritual director. The confessor was described as the concrete mediator and visible face of God's mercy: "*The decisive factor for a true confession is that there is mercy, the unconditional embrace of the person*" (Santa Rosa). Priests describe this moment as a divine act communicated through them: "*Above all, it is a question of Grace; confession is a sacrament, and so it is ultimately a divine initiative*" (Sacristán). Their main task is to invoke the Holy Spirit and become an instrument of God's forgiveness: "*You have to be praying throughout the confession so that the Holy Spirit acts with actual grace on the penitent*" (Corazón). Through confession, they strive to convey to each person "*that God loves them as they are, and that their sin has a solution*" (Piedra).

While, for believers, the forgiveness comes from God, the human presence of the confessor could be very important for the penitent to feel it: "*God has made the sacrament through human relationship, because you need ears to listen to you, eyes to look at you, and arms to embrace you*" (Felicidad). The certainty of forgiveness is uniquely tied to the words of the priest: "*You can only be certain of God's forgiveness when the priest, as God's instrument, says, 'I absolve you from your sins'*" (Juan Pablo II).

Some priests recognize that, although God's forgiveness is not limited by their mediation, the way they conduct confession can profoundly affect the penitent's psychological experience. They strive to create a welcoming, nonjudgmental space: "*My goal is to build a bond where people feel safe and free to share without fear of judgment*" (Independencia). Kindness, understanding, and even humor are seen as helpful: "*I think it's very important that people always see the confessor as kind, understanding, welcoming, positive, optimistic*" (Cercanía). Conversely, negative experiences, such as a severe penance or a distant manner, can potentially thwart the experience of grace and discourage returning to confession: "*If someone leaves the last confession or conversation with a priest feeling uncomfortable... it will obviously be harder for them to return*" (Independencia).

Finally, all interviewed priests acknowledged that their own experiences as penitents shape how they confess and help them adopt an empathic stance: "*Being a penitent yourself helps you try to be like that when you administer confession*" (Cercanía). Taken together, these accounts highlight confession as a fundamentally relational and lived experience, embedded in ongoing interpersonal dynamics. Building on this relational foundation, the following theme explores how these experiences unfold as dynamic processes of reconciliation over time.

## Theme 2. Dynamic movements of reconciliation

Various movements emerge during the process of experiencing reconciliation with God, each corresponding to different phases of confession. These dynamic shifts include moments of self-examination and contrition, the courageous act of confessing, the receiving of DF, and finally, the experience of inner peace and renewed relationship with God.

**Disposition and preparation: The beginning of the process.** The experience of sacramental reconciliation in the Catholic tradition begins long before entering the confessional. Inner preparation—shaped by the examination of conscience and previous experiences—sets the starting point for the journey toward reconciliation. As one priest explained: "*Well, it's about how you prepare yourself. That is, the examination of conscience, sorrow for sins, the purpose of amendment, the confession of sins, and then fulfilling the penance*" (Sacristán). Both personal disposition and faith are considered crucial conditions for fully experiencing the sacrament: "*I would say that for people who approach this sacrament with faith, it does them a lot of good that it be with faith, and that personal dispositions are necessary; they benefit greatly from it*" (Independencia).

**Repentance: Catalyst for change.** The sincere recognition of one's own sin and genuine repentance emerge as essential requirements and authentic drivers of personal transformation. As reflected by one priest, the experience of DF is not withheld by God, but requires repentance as the necessary step by which individuals open themselves to receive absolution: "*But sometimes, let's say, God's forgiveness always, always passes through repentance. And when a person cannot receive absolution, it's not because God doesn't want to forgive them, but because they themselves are unable to express repentance. That's different.*" (Lic).

**Verbal confession: From the interior to the shared.** The verbal expression of sins before the confessor was described as marking a decisive transition, as guilt moves from being an internal reality to becoming something shared and acknowledged. As one priest described: "*In confession* [...] *what the person has to do to receive forgiveness is first to recount their own sins, to put them outside themselves.* [...] *They are going to receive a forgiveness that they do not invent themselves; it is an official forgiveness from a priest*" (17 Junio). This element, which emerged from the interview analysis, points to an understanding that the priest, as a representative of God, plays a crucial role in the process of experiencing reconciliation with God during Catholic confession.

**The confessor's mediation: Accompaniment and acceptance.** The confessor was described as having an important mediating influence in the journey of reconciliation. Serving as both a reflection of God and of humanity, the confessor intends to help the penitent feel that God is looking at them and embracing them—something that is especially profound when the penitent seeks God through the confessor. As one priest described: "*I came with a deep need to experience that embrace of God*" (Santa Rosa).

**Penance: Reflection and integration.** Although penance can be understood as punishment, priests prefer to present it as an invitation to reflection and personal integration. Priests generally see penance as important for absolution: "*The penance is important because it helps to repair, at least in some way, the harm caused by sin*" (Lic). Many propose concrete acts after confession, sometimes as necessary for absolution: "*If someone has committed a serious sin, you can't just say 'three Hail Marys and that's it.' In those cases, I might suggest they do something more significant, like a week of silent retreat or some concrete act of reparation*" (Santa Rosa).

Others believe penance can be understood as the pain caused by sin itself: "*the shame you may feel in confession is already part of the penance, which helps us realize that we have a voice of God within us*" (Piedra). Another priest emphasizes that penance should be rooted in personal reflection rather than as a human requirement for God's forgiveness, which is ultimately given by divine will (grace): "*As a rule, I don't assign penance, because it seems to me that you're doubting God's forgiveness. But I always invite them to reflect on certain things, especially if there's a more serious issue*" (Independencia).

**Absolution: Grace, peace, and a new beginning.** Sacramental absolution was sometimes described as a moment of grace, relief, and renewal, bringing together the self, the confessor, God, others who may have been affected by the personal sin, and the community. As one priest expressed: "*When you come to confession, especially if you've*

experienced it before or for the first time, you realize that God's forgiveness always comes, it always reaches everything and fills you with deep peace and tranquility" (Cercanía). Another observed, "*People always leave better, generally happier*" (Independencia). One priest summarized, "*Grace is received through confession, and it arrives in such a way that many people regain peace, serenity, and feel welcomed*" (Lic).

**Culmination: Reconciliation and renewal.**  Ultimately, some of the interviewed priests show that the sacrament of confession is experienced as a process of reconciliation and restoration of the relationship with God. This journey is described in terms of healing, joy, embrace, conversion, and return to communion with God: "*On the one hand, God's forgiveness is absolute—he always forgives you. And the joy of the penitent when they are reconciled with God...*" (Cercanía). God was consistently described as ready to welcome the penitent into reconciliation, which in turn makes reconciliation with oneself possible: "*It is God who forgives you. And from there, from the healing and the love and mercy that he pours out, you can reconcile with yourself, but without God's strength, you will never do it*" (Piedra). Confession can act as a catalyst for this process: "*It is about taking a step forward in the relationship with God. That, in confession, is a great impetu*s" (Corazón). At the same time, reconciliation is not limited to the single act of confessing, but must be integrated into a broader spiritual experience: "*I believe that what is fundamental is that you live confession within an experience of faith and spirituality*" (Independencia).

While these dynamic movements of reconciliation illustrate how the process ideally unfolds, participants also pointed to conditions that can hinder or facilitate these trajectories. The next theme therefore examines the perceived barriers and facilitators shaping these experiences.

### Theme 3. Perceived barriers and facilitators

**Barriers of reconciliation.**  The first type of barriers identified are personal, related to guilt, self-image, scrupulosity, and shame. Persistent guilt can become a major obstacle to experiencing DF: "*Excessive guilt is of no help, because people feel very bad, and then, even once they stabilize, the issue comes up again; there is no real inner healing*" (Independencia). The role of guilt becomes especially problematic when it leads to scrupulosity—an obsessive psychological tendency to confess even previously absolved sins—which can create a cycle of repeated guilt: "*They never feel forgiven* [...] *they blame themselves a lot* [...] *they keep confessing the same sin*" (Felicidad).

Guilt appears to take on a similar role to shame when it relates to its effect on one's self-image: "*Guilt is when you don't feel worthy, you can't look at yourself with tenderness. So, what happens? Guilt buries you in some way*" (Felicidad). And shame itself could also be a significant obstacle to experiencing reconciliation with God: "*if you don't overcome it, it can block you. Because of shame, people may not verbalize everything and then leave feeling uncomfortable*" (Independencia); and can also hinder self-forgiveness: "*Shame does not help you experience God's forgiveness. And of course, if God doesn't forgive you, then you can't forgive yourself*" (Felicidad).

Beyond internal struggles, relational and contextual factors shape the confessional experience. Negative experiences with the confessor—such as a lack of empathy, rushing, or not feeling truly heard—can discourage returning to confession: "*If you're in a hurry or don't feel truly heard, it's much harder to open up*" (Cercanía). Negative previous experiences may even distort one's image of God: "*God is Father, but the personal experience someone had of a father may have been one who abandoned them...*" (Juan Pablo II).

Whether confession is face-to-face or anonymous, and whether the confessor is personally known or a stranger, can also influence the comfort and openness of the penitent. Some priests feel it is better not to know the confessor, while others value personal connection: "*It was much better that the confessor knew me… that was the great discovery*" (Sacristán). For some, routine confession can become merely habitual: "*Sometimes people confess so often it becomes just a routine, and they lose the sense of what's really happening*" (Santa Rosa). Cultural and institutional barriers—such as insufficient catechesis or lack of access—may further limit participation: "*Low frequency of confession can be the result of insufficient religious education and catechesis*" (Corazón).

**Facilitators of reconciliation.** At a personal level, while persistent or deep guilt and shame can become obstacles—as discussed above—some priests also noted their potential positive role when they catalyze recognizing one's sins: "*Guilt is at the foundation of the experience of forgiveness*" (Corazón). As another priest explained, "*The clearer people are about their personal guilt, the better their confession and sacramental experience will be*" (Sacristán). "*Shame, on the one hand, helps you recognize the seriousness* [of the sin]" (Independencia). And once shame is overcome, it can be experienced as a great relief: "*It's liberating. The thing is, it's hard, it's hard because it requires trust—these are generally things you don't tell anyone*" (Felicidad).

At the relational level, the presence and availability of the confessor appear to be decisive: "*A priest who is always there with the green light on… that gives people a sense of security*" (17 Junio). Once present, the confessor's gestures—such as a warm welcome, a smile, use of humor or encouraging words—can ease anxiety and promote openness in confession. Also, reminding penitents of God's unconditional mercy and forgiveness fosters trust and hope: "*Reminding them of Gospel passages where Jesus forgives everyone—so you won't be an exception—makes a big difference*" (Sacristán).

In addition, attending religious services often predisposes people to confession: "*Many times after Mass or adoration, people feel the need and ask to go to confession*" (Santa Rosa). While regular confession can become routine, some priests stress its value, noting that the ideal frequency is personal: "*I always say it depends, but once a month is not a bad idea. Even if people feel they have no serious sins, I tell them a regular tune-up keeps everything running smoothly*" (Piedra).

Importantly, these barriers and facilitators structure the process externally but they are also intertwined with participants' internal experiences. To better understand this subjective dimension, the next theme focuses on the cognitive and emotional aspects of DF.

### Theme 4. Cognition and emotion in the experience of DF

**Cognitive and doctrinal dimension.** Priests distinguish between rational certainty of DF and emotional experience: "*I know—there is a cognitive, rational dimension that I know I have been forgiven. It's another thing if God gives me a particular grace to feel his forgiveness*" (Sacristán). One priest emphasized that reconciliation with God should not depend solely on feelings, but also on rational understanding and the will to repent: "*We must try to look at this with our minds. The pain should come more from reason and will—I know this has offended God, and if I feel it, it will help me ask for forgiveness*" (Cercanía). In this sense, rationality can lead to the experience of reconciliation, while emotions remain less controllable: "*What makes you free is having intelligence and will. Feeling is something you don't control*" (Cercanía).

**Emotional and experiential dimension.** The emotional experience of DF can be highly significant, especially when related to intense or painful feelings of sin and confession. "*Confession helps you grow in grace* [...] *if you go in need, you experience it more strongly. It is a very satisfying experience because you truly come to feel that forgiveness*" (Felicidad). Some priests describe how the experience of God's mercy, through invoking the Holy Spirit and God's word, can change the heart: "*The experience of God's mercy, through the invocation of the Holy Spirit and the Word of God, changes the heart*" (Corazón). For some priests, these experiences are so powerful that they shape their very own vocation: "*My own experience of forgiveness as a priest in confession—even in my vocational journey—confession has been a very important point of discovering God and deepening my priestly vocation*" (Sacristán). Priests note that DF may be experienced with intense emotions or, at times, without any strong feelings; in both cases, forgiveness is still granted: "*There are graces that the Lord may grant—sometimes experienced as joy, as tears, as inner peace—but sometimes not, and yet you still know that the Lord has forgiven yo*u" (Sacristán).

Taken together, the cognitive and emotional dimensions reveal how participants make sense of and engage with DF at an individual level. The final theme integrates these insights with the relational and dynamic aspects previously described, offering a more comprehensive understanding of the phenomenon.

## Theme 5. Integrative axis: The relational fabric and the dynamism of DF

Bringing together the relational, processual, and experiential dimensions identified in the previous themes, this integrative axis highlights how these elements converge into a coherent understanding of DF. The experience of DF in the Catholic tradition was not described by participants as consisting of separate processes or dimensions, but rather as a fabric in which the different axes interrelate and mutually influence one another. Axis 1 (the relational fabric) highlights how participants described the self, God, others, sin, the community, and the confessor as connected in an organic network. As one priest put it, "*we cannot conceive of ourselves outside of relationship, because if we are made in the image of God, then we are relationship, just as God is relationship in his very nature*" (Felicidad). In this way, personal identity was described as being mediated by the gaze and relationship with others: "*You cannot look at yourself with tenderness except through the gaze of another who looks at you with tenderness*" (Felicidad).

Within this fabric, the movements of forgiveness (Axis 2) were described as unfolding in dynamic and reciprocal ways. The process of experiencing reconciliation was often described as beginning with a call from God, who "*always needs a response from me, even if it may often seem very simple or very small*" (Lic). This response may be either facilitated or hindered by facilitators and barriers (Axis 3), such as the community, liturgical moments, or the confessor's attitude. Difficulties increase especially when sin has affected others: "*people have more difficulty experiencing God's forgiveness in sins that have affected other people*" (Cercanía). Sometimes, it is the confessor who helps the penitent become aware of the impact on others, inviting them to consider "*not only their actions, but how those actions affected others, to promote true contrition*" (Sacristán).

In this process, one's image of God could reflect one's self-image and shape the experience of forgiveness: "*The person identifies with what they have done. It's a vision of God, that you think God looks at you that way*" (Felicidad). The welcoming climate created by the confessor was referred to as a key—a distant attitude can make the encounter more difficult, whereas "*creating a context of dialogue with God, of God's presence, helps a lot*" (Cercanía). Finally, Axis 4 (the experience of forgiveness) was described as taking concrete shape in various ways: sometimes it is lived as an intense emotional experience, while at other times it is simply a rational certainty upon hearing the words of absolution. For these participants, the assurance of having been forgiven often comes through the words of the priest who grants absolution in the name of God, and emotion is not required to receive DF. Thus, confession appears in participants' accounts as a relational and dynamic process in which all four axes—the relational fabric, the movements, the facilitators/barriers, and the subjective experience of forgiveness—intertwine to create a unique and ever-evolving experience.

## Discussion

This study draws from the narratives of Spanish-speaking priests to explore their experiences of DF in Catholic confession. Using a theory-informed framework analysis combining inductive coding with iterative matrix-based synthesis, these experiences (in both confession and its surrounding context) were analyzed and subsequently discussed in dialogue with the Christian-sensitive model of experiencing reconciliation with God after recognition of personal sin [7,12,15,39,40]. The findings provide a contextualized understanding of how DF tends to unfold in this setting, integrating relational, emotional, and cognitive dimensions. Catholic confession in this sample emerges as a dynamic and relational process in which seeking and receiving DF cannot be reduced to individual experience, but rather appears to involve, in participants' accounts, an ongoing interaction between God, the self, others affected, the community, and—distinctively compared to many other traditions [22]—the confessor. Our analysis identified four central axes: 1) the relational dimension and its elements; 2) the dynamic movement between repentance and absolution; 3) the role of facilitators and barriers; and 4) the interplay of rational and emotional aspects of DF. These axes highlight confession as an integrative process, with the confessor described as playing a central mediating role. We unpack these findings in light of Christian-sensitive [7] and faith-neutral [3] models of DF, points of connections with contemporary reflections on the role of the confessor, and potential implications of the findings.

Our results confirm the centrality of the main elements of Cook and Cowden's [7] broadly ecumenical relational spirituality model of personal sin within the Catholic confession experience, as reflected in our first thematic axis: the self, God, personal sin, and others. At the same time, our findings suggest relational particularities within this context that may add complexity to the model. Specifically, the interviews suggest that, within this context, the confessor could be considered by priests as a unique and irreplaceable figure in the process of experiencing reconciliation with God within the Catholic tradition: he not only represents the religious community, but during confession, is described by participants as the sacramental mediator who is thought to embody divine authority and mercy. When the priest says "I absolve you" in every confession, he is understood within Catholic doctrine and by priest participants, as sacramentally identifying with Christ and enacting DF, which is experienced as allowing the penitent to experience reconciliation with God [41,42]. Thus, the act of mercy and DF may be experienced as both divine and human, carried out by a priest who is also a sinner and penitent. This understanding of DF as a relational experience is supported by contemporary confessor voices in literature. For example, Fernández Castiella [43] refers to confession as an "epiphany of relationship," emphasizing that "we are in an eminently relational sphere" where only in this context can true mediation of mercy occur. This view is also closely related to Wise's [27] findings on confession from the penitent's perspective, where it is perceived as an organic whole in which the various elements and stages are felt as united [18].

Within the explained relational spirituality framework, the priest as confessor appears to mediate the reciprocal movements between penitent and God, described in our second axis of results, resonating with Cowden et al.'s [14] model of experiencing reconciliation with God. By embodying both humanity and the sacramental presence of Christ during the moment of confession, the confessor may function as a bridge between the psychological and theological dimensions of forgiveness. The confessor also participates in shaping facilitators and barriers to the psychological experience of DF (as highlighted in the third thematic axis). Notably, a warm, non-judgmental climate during confession emerges in participants' accounts as especially beneficial—an insight echoed by contemporary reflections on the confessor's role [24] and previously shown from the perspective of lay penitents [27]. The relevance recognized by the priests in the interviews about these human elements (such as a welcoming attitude) for the experience of DF in confession supports participants' conceptualization of the confessor in this dual role: as God's representative, but also a human being like me. This relational and pastoral context may shape the penitent's psychological experience of DF, ranging from deeply emotional and intense to more rational and cognitive, as illustrated in our fourth thematic axis.

Based on the data and analysis reported in this study, the confessor—as a singular figure—does not seem to fit neatly into any of the proposed vertices of Cook and Cowden's [7] psychologically-based relational spirituality model of personal sin (Fig 1) when it comes to experiencing reconciliation with God from the priests' perspective within the context of Catholic confession. The confessor may not be fully reducible to the "others" axis in the model, as penitent priests are aware that the confessor acts *in persona Christi*; nor could the confessor be completely identified with the "God" axis, since a priest is not God. Another possibility would be to position the confessor as a mediating figure between the "self" and "God" nodes. However, participants were aware that Catholic doctrine allows for direct access to DF without priestly mediation in exceptional circumstances, such as in danger of death [17]. As a result, they may also see the possibility of direct reconciliation with God as an indication that priestly mediation between the "self" and God is not always experienced as necessary. From the distilled perspectives of the small sample of Catholic priests included in this study, we tentatively suggest that the confessor may be conceptualized as a context-specific relational element that could be represented as a distinct fourth vertex within the model (see Fig 2) in this specific context. This proposal should be understood as a hypothesis-generating extension requiring further empirical examination across contexts. Accordingly, this interpretation should not be taken as a structural modification of the model, but rather as a contextually grounded elaboration within some sacramental Catholic settings.

Based on our tentative proposal for an expanded relational spirituality model of personal sin for the Catholic tradition, the penitent's experience of reconciliation with God may be conceptualized as following either a direct route

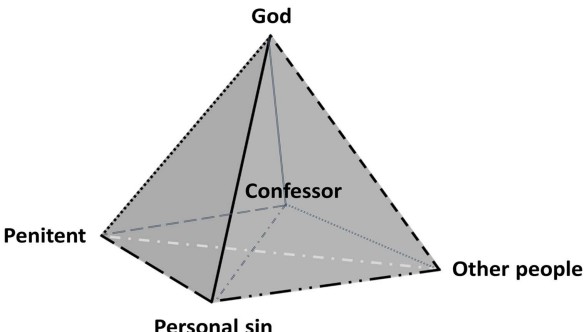

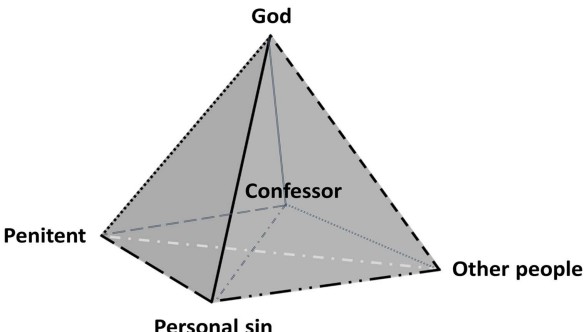

**Fig 2. A Contextually grounded, exploratory extension of the Christian-sensitive relational spirituality model of personal sin within Catholic confession.** This figure illustrates a context-specific and exploratory extension of Cook and Cowden's [7] relational spirituality model of personal sin based on the perspective of Catholic priests. As a tentative proposal, it requires further empirical validation and does not necessarily represent the views of lay Catholics. In this model proposal, the penitent is used in place of "self" and the confessor is provisionally represented as a distinct relational element that may be conceptualized as an additional vertex at the base of the pyramid, directly linked to the penitent, God, personal sin, and others as shown by the edges: 1) Confessor–Penitent: The confessor may be experienced as an active companion who welcomes the penitent's vulnerability, helps them overcome shame, and facilitates the verbalization of sin, potentially supporting self-knowledge and openness to grace. 2) Confessor–Sin: Verbal confession before the priest may contribute to transforming sin into an objective and shared reality, allowing guilt to be reframed not only as a source of shame but as a starting point for change and reconciliation. 3) Confessor–God: The priest is understood within Catholic doctrine as acting *in persona Christi* at the moment of absolution, which may make DF tangibly present and providing objective certainty of reconciliation that may go beyond personal feelings. 4) Confessor–Others: The confessor may function as a link between the penitent and the religious community, representing its welcome and testimony, with existing bonds potentially reinforcing the experience of belonging and integration. The confessor may also help the penitent reflect on how their actions might have impacted others.

(Penitent–God) or an indirect route (Penitent–Confessor–God). The confessor may likewise be involved in mediating (or not) the penitent's relationship with 'Others' and the 'Personal sin' nodes, corresponding to indirect or direct relational pathways. These configurations may generate complex interactions that warrant further study, likely shaped by individual differences in experiential and doctrinal dimensions of spirituality. For example, some penitents may feel able to confess directly to God, whereas others may experience DF more fully through the priest's discernment of the specific sin, his words of absolution, and the assignment of penance, which may involve acts of reparation toward others.

Importantly, the possibility of including a potential new node labeled "confessor" may apply mainly to priests rather than to the broader Catholic population, influenced by factors such as participation in confession and theological formation. Although confessing grave sins at least once a year is a canonical obligation [17], observance varies: priests tend to receive the sacrament more frequently, whereas only about 20% of lay Catholics do so [44,45]. Limited participation may lead many lay Catholics to psychologically experience DF directly, positioning the confessor as part of the "other" node (e.g., clergy or community) rather than as a distinct essential element. Moreover, priests' understanding that the confessor acts in persona Christi may strengthen their association of the confessor with God, while doctrinal knowledge and experiences within the Catholic tradition could also shape their perceptions [12]. Future research should examine how roles, training, and experience influence perceptions of the confessor's position within the relational spirituality model of personal sin, as well as whether discrepancies between Catholic doctrine about the confessor and perceptions of the confessor among lay Catholics might impact experiences of DF.

Our findings can also be understood in light of Fincham and May's [3] model of DF seeking. Recognition of a sin—often prompted by guilt, shame, or scrupulosity—reflects the model's initial stage. The decision to seek DF through sacramental confession, appears clearly in the accounts of Catholic priests (although, as noted above, it may not necessarily be experienced in the same way by lay Catholics) and is shaped by internal dispositions (e.g., spiritual formation, God image) and external influences (e.g., parish context, relationship with the confessor), paralleling the model's emphasis on motivational

and relational influences. The decision point concerning conditionality also maps onto Catholic practice: confession appears to provide a clear behavioral condition through which DF is commonly experienced by priests. At the same time, our data could expand and contextualize the model in key ways. First, priests describe the process as relational and dynamic rather than individual and linear, involving recursive movements across preparation, confession, and post-confessional reflection. Second, the adaptive endpoint is not only the perception of DF but often what participants described as a deeper sense of reconciliation or restored communion with God—a relational and sacramental restoration catalyzed by confession but unfolding over time. This aligns closely with Cook and Cowden's [7] relational spirituality model, which frames a restored relationship with God as the ultimate goal of the process [14,15]. Interestingly, within this arc, conditionality is reframed: while sacramental confession is seen by participants as necessary, priests emphasize that God's mercy is ultimately unconditional, awaiting the penitent's openness.

Finally, drawing on both models, the present findings provide insight into how confession may function at both the cognitive and emotional levels of experiencing DF. The sacramental ritual appears primarily to confer cognitive certainty of being forgiven, which in some cases may facilitate the emotional realization of forgiveness; however, such affective experience is neither theologically required nor consistently achieved. Emotional awareness may also arise through the relational dynamics of the confessional encounter, particularly the confessor's empathetic, accepting, and nonjudgmental presence, which may deepen the penitent's sense of reconciliation. Thus, confession appears to involve elements that address cognitive, emotional, and integrative dimensions, although the full alignment between knowing and feeling forgiven is not always attained. This interpretation contextualizes existing DF models within Catholic practice, showing that Fincham and May's [3] cognitive–affective distinction and Cook and Cowden's [7] relational framework appear to find specific expression in the interplay between sacramental assurance and the confessor–penitent relationship.

Our sample of Spanish-speaking priests from various countries provides a valuable lens into the role of culture in the experience of DF. Although participants did not explicitly reflect on their Spanish-speaking background in the interviews, their narratives consistently emphasized the relational and communal nature of DF. This emphasis resonates with research characterizing Spanish-speaking comunities [46] and Hispanic spirituality as especially communal and relational [29,30]. While the role of the priest as confessor and ecclesial mediator is central to Catholicism broadly, our findings suggest that, in Hispanic contexts, this role may carry heightened relational and communal significance—further strengthening the bond between the penitent, the wider faith community, the confessor, and ultimately, God. As our findings indicate, DF in Spanish-speaking Catholic contexts as reflected in this sample is fundamentally understood as a relational phenomenon, inseparable from its communal context—a dynamic that carries significant implications for both theory and practice, as explored below.

## Implications

These findings have implications for both psychological theory and pastoral practice. Our findings build on Cook and Cowden's [7] model of reconciliation with God and suggest possible contextual extensions when considering the perspective of Catholic priests, in which the confessor may be conceptualized as a distinct and irreducible node within the relational dynamic, a suggestion that requires further empirical support. Additionally, our findings contextualize elements of Fincham and May's [3] model, such as the role of conditionality in DF within Catholicism. These insights can inform future frameworks on DF, highlighting the role of relational mediation and ritual. Practically, the results suggest that the priest's empathetic and nonjudgmental approach has the potential to support the penitent's experience of DF, which has relevance for clergy formation and pastoral care in line with current pastoral proposals [24,43]. Moreover, the study highlights the value of qualitative methods in revealing culturally embedded experiences of DF, and calls for greater intercultural awareness in research and pastoral training, particularly in increasingly diverse Catholic contexts.

## Limitations

This study has some limitations. The sample consisted of experienced Spanish-speaking Catholic priests, offering rich, theologically grounded insights but representing a relatively homogeneous group, with most participants based in Spain, which may have influenced the scope of the findings. This limits perspective diversity and may constrain the transferability of findings to other contexts (e.g., laity or younger, less experienced confessors). The results should therefore be understood as specific to Catholic priests in Spanish-speaking settings under the sacramental rite of confession, rather than generalized to lay Catholics or other Christian or linguistic contexts. The proposed preliminary expansion of Cook and Cowden's [7] model may likewise reflect this population's experience, potentially overlooking dimensions more characteristic of less theologically formed individuals. Responses may also have been shaped—consciously or not—by social desirability, given the sensitive nature of the topic and participants' institutional roles. Internal diversity (e.g., age, country, or years of ministry) introduced variation, though it also enriched the analysis. Finally, while the study considered related practices such as spiritual direction, its primary focus on confession may limit how reconciliation is understood in other moments in the DF-seeking process. Moreover, the cross-sectional design restricts insight into how this experience of reconciliation may evolve over time.

## Future research directions

To deepen understanding of DF in the Catholic context, future research is needed. Regarding theoretical models of DF, first, further qualitative and quantitative work examining how Catholics (both laypeople and priests) conceptualize and situate the role of the confessor within the Christian-sensitive relational spirituality model of personal sin [7], could test and assess the scope of our proposal of the confessor as a potential additional node in the model. Second, mapping the sacramental process step by step, and analyzing this processual experience from contrition to confession, penance, and absolution, would clarify the applicability of Fincham and May's [3] framework for seeking DF. To gain a deeper understanding of the role of confession in the experience of DF and other psychological variables, dyadic studies including both priests and penitents would be especially valuable for capturing the relational dynamics of confession and offering deeper insight into how reconciliation is co-constructed. Finally, further longitudinal studies (e.g., testing key variables before and after confession) might identify facilitators and barriers to experiencing reconciliation with God after a personal sin in Catholics, as well as clarify the effects of confession and the experience of DF on well-being.

## Conclusion

In this qualitative study of priests from four Spanish-speaking countries, we explored experiences of DF in Catholic confession. The findings suggest that participants portrayed confession as a dynamic, relational, and meaning-making process shaped by the interplay of self, sin, God, others, and—distinctively—the confessor, integrating both universal and culturally specific dimensions of reconciliation with God. By proposing a preliminary exploratory expansion of Cook and Cowden's [7] model of DF for the Catholic context and situating our results in dialogue with Fincham and May's [3] faith-neutral framework, this study contributes to the conceptual development of the field and to a deeper understanding of the psychological processes underlying the experience of DF. The confessor's empathy and the communal setting of confession appear especially salient in participants' accounts of forgiveness-related experiences and psychological outcomes, particularly within Spanish-speaking cultures where relationality is highly valued [30,46]. While limited to priests, this study offers an initial, culturally grounded step toward an integrated psychological understanding of DF. Future research should include lay perspectives and employ dyadic or longitudinal designs to further clarify the role of confession as both a ritual and a transformative relational space. Overall, the results underscore the value of qualitative inquiry for capturing the complexity of spiritual experience and call for continued dialogue between psychological theory, culture, and religious practice.

## Supporting information

**S1 Table. COREQ 32-Item checklist for the present study.** The Consolidated Criteria for Reporting Qualitative Research (COREQ) is a 32-item checklist designed to improve transparency and completeness in reporting qualitative studies based on interviews and focus groups. Although the information corresponding to these criteria is integrated throughout the manuscript in the relevant methodological and analytic sections, all COREQ items are compiled here in a single supplementary table to provide a clear and comprehensive overview of how each criterion was addressed in the design, conduct, analysis, and reporting of the present study.
(DOCX)

**S2 Table. Themes, subthemes, and associated codes (English labels with Spanish terms in parentheses).** Codes were initially informed by the Reconciliation with God framework (Cook & Cowden, 2025) and then expanded through inductive analysis beyond this framework. Coding proceeded bottom-up in three hierarchical levels: Level 3 (most specific), Level 2 (broader), and Level 1 (most general). English labels are shown with their original Spanish terms in parentheses (e.g., Repentance (*Arrepentimiento*)). Codes were first developed at Level 3, then clustered into broader Level 2 and Level 1 codes, which were subsequently grouped into categories, subthemes, and overarching themes. The complete coding matrix is provided in the main manuscript as a referenced supplementary file.
(DOCX)

**S1 Checklist. Inclusivity in global research.**
(DOCX)

## Acknowledgments

We thank Drs. María Arantzamendi and Alazne Belar for their methodological guidance throughout the development of this study, which greatly strengthened its design and execution. We also thank Dr. Xavier Bringué for his valuable collaboration in data collection. The authors extend their heartfelt thanks to the participants for their active participation.

## Author contributions

**Conceptualization:** Carmen Callizo, Martiño Rodríguez-González, María Calatrava.

**Data curation:** Carmen Callizo, Martiño Rodríguez-González.

**Formal analysis:** Carmen Callizo, María Calatrava.

**Funding acquisition:** Martiño Rodríguez-González, María Calatrava.

**Investigation:** Carmen Callizo, Martiño Rodríguez-González.

**Methodology:** Carmen Callizo.

**Project administration:** Carmen Callizo, María Pilar Martínez-Díaz.

**Resources:** Carmen Callizo.

**Software:** Carmen Callizo.

**Supervision:** Carmen Callizo, Martiño Rodríguez-González, María Calatrava, Richard G. Cowden.

**Validation:** Carmen Callizo.

**Visualization:** Carmen Callizo.

**Writing – original draft:** Carmen Callizo.

**Writing – review & editing:** Martiño Rodríguez-González, María Calatrava, María Pilar Martínez-Díaz, Kaye Cook, Richard G. Cowden.

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
