## [Decision Letter · Decision Letter 0]

18 Mar 2026

PONE-D-25-62289Mediating Reconciliation With God: Exploring Divine Forgiveness Experiences During Confession Among Catholic Priests From Four Spanish-Speaking CountriesPLOS One

Dear Dr. Rodríguez-González,

Thank you for submitting your manuscript to PLOS ONE. After careful consideration, we feel that it has merit but has yet to fully meet PLOS ONE’s publication criteria as it currently stands. Therefore, we invite you to submit a revised version of the manuscript that addresses the points raised during the review process.

We look forward to receiving your revised manuscript.

Kind regards,

Rafael Galvão de Almeida, PhD.

Academic Editor

PLOS One

Journal Requirements:

5. Please amend your authorship list in your manuscript file to include author Martiño Rodríguez-González, Carmen Callizo, Maria Calatrava, Maria Pilar Martínez-Díaz, Maria Pilar Martínez-Díaz, Richard G. Cowden.

6. Please amend the manuscript submission data (via Edit Submission) to include author Rodríguez-González, M., Callizo, C., Calatrava, M., Martínez Díaz, M.ª P., Cook, K., & Cowden, R. G.

7. Your abstract cannot contain citations. Please only include citations in the body text of the manuscript, and ensure that they remain in ascending numerical order on first mention.

Reviewers' comments:

Reviewer's Responses to Questions

**Comments to the Author**

1. Is the manuscript technically sound, and do the data support the conclusions?

Reviewer #1: Yes

Reviewer #2: Yes

2. Has the statistical analysis been performed appropriately and rigorously? 

Reviewer #1: N/A

Reviewer #2: N/A

3. Have the authors made all data underlying the findings in their manuscript fully available?

Reviewer #1: Yes

Reviewer #2: Yes

4. Is the manuscript presented in an intelligible fashion and written in standard English?

Reviewer #1: Yes

Reviewer #2: Yes

5. Review Comments to the Author

Reviewer #1: This manuscript presents a carefully conducted qualitative study exploring Catholic priests’ experiences and understandings of divine forgiveness (DF) within sacramental confession. The topic is original, theoretically engaged, and timely within the emerging psychological literature on divine forgiveness.

One of the strongest aspects of the manuscript is the rigorous and sustained engagement with the two central theoretical models (Fincham & May; Cook & Cowden). The authors not only present these models clearly and accurately in the Introduction, but also maintain them as analytic reference points throughout the Results and Discussion. The models are neither superficially cited nor merely decorative; rather, they structure the analytic logic of the study and are revisited critically in light of the data. This theoretical coherence significantly strengthens the manuscript and distinguishes it from purely descriptive qualitative work.

The study is methodologically transparent, the analytic process is clearly described, and the integration with existing theoretical models is thoughtful. The manuscript has clear potential for publication. However, several conceptual and interpretive issues require clarification before it can be considered suitable for publication in a general scientific journal such as PLOS ONE.

My comments focus on strengthening theoretical precision, interpretive restraint, and epistemological clarity.

1. Scope of Theoretical Claims

The manuscript proposes that the confessor may constitute a distinct “fourth vertex” in the relational spirituality model of personal sin. While this is a compelling interpretive insight, the current formulation risks overstating the implications of findings derived from a small and relatively homogeneous sample (N = 10 experienced male priests, predominantly from Spain).

I recommend moderating the strength of this claim. Rather than suggesting structural modification of the model, the argument would be more appropriately framed as a context-specific elaboration within sacramental Catholic settings, and a hypothesis-generating extension requiring further empirical examination. The contribution remains valuable, but the language should reflect its exploratory status.

2. Distinction Between Psychological Description and Theological Normativity

The manuscript generally maintains a psychological lens; however, in several passages—particularly in the Results and Discussion—the wording occasionally moves very close to doctrinal formulation rather than analytic description.

For example: In the Results, statements such as “You can only be certain of God’s forgiveness when the priest… says ‘I absolve you’” are presented with minimal analytic framing. While clearly derived from participant quotations, the surrounding text could more explicitly indicate that this reflects a participant belief structure rather than an ontological claim about the nature of forgiveness. In the Discussion, the description of the confessor as one who “sacramentally identifies with Christ and enacts DF” reads, at points, as a theological assertion rather than as an account of how priests conceptualize their role. A small shift in phrasing (e.g., “is understood by participants as…”) would maintain scientific neutrality. Similarly, when describing absolution as the moment in which the penitent “is reunited with God,” the manuscript could clarify that this is a phenomenological or doctrinally mediated interpretation rather than an empirically demonstrable event.

None of these instances undermine the study. However, because PLOS ONE is a general scientific journal, maintaining a consistently descriptive psychological register—especially when dealing with sacramental theology—would enhance epistemological clarity. The suggested revisions require only slight adjustments in phrasing, not substantive reanalysis.

3. Cultural Framing and Representativeness

Although the study includes participants from four Spanish-speaking countries, six of the ten participants are based in Spain. At several points, the manuscript refers broadly to “Spanish-speaking contexts.” I recommend slightly narrowing this phrasing or explicitly acknowledging the uneven distribution of participants across countries to avoid unintended generalization.

4. Use of AI as Analytic Aid

The transparency regarding the use of ChatGPT as an auxiliary analytic tool is commendable. Given the novelty of AI-assisted qualitative analysis, I suggest slightly expanding the clarification that: AI did not generate primary codes or interpretive structures, all analytic decisions remained with the research team, and the tool was used solely for robustness checks.

This would pre-empt potential methodological concerns.

Overall Evaluation

This is a conceptually strong, theoretically coherent, and methodologically careful qualitative study that meaningfully advances empirical research on divine forgiveness within a sacramental Catholic context. The sustained integration of contemporary theoretical models is particularly noteworthy.

With modest refinements to theoretical framing and epistemological positioning, the manuscript would represent a valuable contribution to the field.

I encourage revision along the lines suggested above.

Reviewer #2: Review of Manuscript PONE-D-25-62289: "Mediating Reconciliation With God: Exploring Divine Forgiveness Experiences During Confession Among Catholic Priests From Four Spanish-Speaking Countries"

Dear Authors:

Thank you for the opportunity the read and review your article entitled "Mediating Reconciliation With God: Exploring Divine Forgiveness Experiences During Confession Among Catholic Priests From Four Spanish-Speaking Countries." I begin by briefly noting some pronounced strengths of the manuscript.

Strengths

1 – The focus on reports of Catholic priests as confessors is quite novel and innovative. I am not Catholic but the authors provide a deep but “outsider friendly” view of the focal phenomenon. I found the piece deeply engaging.

2 – The Referencing and citations were nearly perfect. The manuscript was lucid and clear … and virtually error free. The piece was well-cited, with heavy reliance on very fresh and recent sources (many from the past 5 years or so).

3 – In terms of qualitative methodology, the piece was rigorous and systematic. The depth of the interviews (mean = 74 minutes) is rare these days. I was also impressed with the use of ChatGPT as an “auxiliary aid” (or audit tool).

In the spirit of precision and constructive criticism, I offer the following recommendations that I hope will be helpful.

1 – On p. 4, line 71, you reference “relational spirituality.” I encourage you to cite Annette Mahoney here (in addition to those already cited). She, I believe, was the innovator of this concept.

2 – Page 7, line 149. In APA 7th Style, when multiple references are cited parenthetically at the end of a statement, these references need to alphabetized by the first authors’ last names. For example, when you make a statement and then cite (Ramirez, 1985; Ruiz, 2005; Gomez, 2003) then Gomez needs to go before Ramirez.

3 – An uncited piece you may want to consider in connection with Confession is:

White, T., et al. (2018). Kept together by faith: Confession and forgiveness among American Catholic and Orthodox Christian families. Marriage & Family Review, 54, 677-692. doi:10.1080/01494929.2018.1469573

4 – On p. 10, you mention a sample of “ten Catholic priests” (line 203), but the numbers given on lines 214-215: “Spain – 6, Peru – 1, Chile – 1, and Mexico – 1” only total 9. Please correct this inconsistency.

5 - Please be careful about “naked” paragraphs—meaning paragraphs without a single reference to the scholarly literature. Occasionally, there may be a need for such a paragraph but I would typically avoid going for too long without anchoring and re-anchoring your work in the scholarly literature. Some “naked” paragraphs occur on pages 13-14 (only one cite on these two pages), and for most of p. 32.

6 – Typically, in qualitative work, the “Results” section is instead labeled as “Findings.” (p. 15)

7 – In the Results/Findings section, I would recommend listing/outlining your major Themes up front (after the first paragraph on p. 15), in the spirit of “tell the reader what you are going to tell them.” Additionally, on a related note, I recommend similarly outlining the sub-themes at the beginning of each theme. Both strategies serve as a reader guide and orientation.

8 – I found the narrative selections to be brief but excellent. I also appreciated the decision to italicize the Priests’ quotes for differentiation and clarity. However, on p. 21 (lines 470-471) you forgot to italicize a priest’s quote, I believe.

9 – One slightly jarring feature of the piece is a lack of a transition sentence or two between the major Themes of the Results/Findings section. I recommend adding transitions between Themes 1 and 2, 2 and 3 (p.23), etc. Please help the reader see the connection between the Themes and why you ordered the presentation of the Themes as you did.

10 – Page 36, line 810, needs to be indented.

I wish you the best as you continue your efforts to publish this manuscript, it is an excellent piece and was a pleasure to read.

6. PLOS authors have the option to publish the peer review history of their article (what does this mean?). If published, this will include your full peer review and any attached files.

Do you want your identity to be public for this peer review? For information about this choice, including consent withdrawal, please see our Privacy Policy.

Reviewer #1:  Yes: Maria Prieto-Ursua

Reviewer #2: No

---

## [Author Response · Author response to Decision Letter 1]

1 Apr 2026

Response to the Editor’s and Reviewers’ Comments

Note: A separate cover letter summarizes the changes made to the manuscript. In this document, we provide a detailed, point-by-point response to each comment raised by the editor and reviewers

Journal Requirements

Response to Journal Requirement #1: We thank the editor for this request. We have carefully adhered to all PLOS ONE formatting requirements and ensured that the manuscript fully complies with the journal’s style guidelines, including those related to file naming.

Response to Journal Requirement #2: The PLOS inclusivity in the global research questionnaire has been completed and uploaded as Supporting Information (S3 Checklist) with the revised submission.

Response to Journal Requirement #3: We have checked that the ethics statement appears exclusively in the Methods section and does not appear elsewhere in the manuscript.

Response to Journal Requirement #4: We thank the editor for this clarification. In line with the journal’s data availability policy, the data have now been made publicly accessible, as specified in the Data Availability Statement.

5. Please amend your authorship list in your manuscript file to include author Martiño Rodríguez-González, Carmen Callizo, Maria Calatrava, Maria Pilar Martínez-Díaz, Maria Pilar Martínez-Díaz, Richard G. Cowden.

6. Please amend the manuscript submission data (via Edit Submission) to include author Rodríguez-González, M., Callizo, C., Calatrava, M., Martínez Díaz, M.ª P., Cook, K., & Cowden, R. G.

Response to Journal Requirement #5&6: We thank the editor for this comment. However, we are unclear about this request, as we have verified that the authors listed in the submission system are consistent with those and their order in the manuscript. We would be grateful for further clarification. We confirm that the authors of the manuscript and the order they should be listed is as follows: Callizo, C., Rodríguez-González, M., Calatrava, M., Martínez-Díaz, M.ª P., Cook, K., & Cowden, R. G.

7. Your abstract cannot contain citations. Please only include citations in the body text of the manuscript, and ensure that they remain in ascending numerical order on first mention.

Response to Journal Requirement #7: We confirm that the abstract does not contain any citations and complies with the journal’s guidelines.

Response to Journal Requirement #8: We have carefully evaluated the references suggested by the reviewers. Following this assessment, we have included two of them, as they are relevant to our work, as detailed in our responses to Reviewer #2.

Response to Journal Requirement #9: We have carefully reviewed and updated the reference list to ensure its completeness and accuracy. Specifically, we have added two references recommended by the reviewer (now References 18 and 40), as well as an additional relevant reference (Reference 46). We have also updated a previously “in press” article to its final published version (Reference 15). No previously included references have been removed. In addition, the entire reference list has been formatted according to the Vancouver style required by PLOS ONE.

Reviewers' comments:

Reviewer's Responses to Questions

Comments to the Author

1. Is the manuscript technically sound, and do the data support the conclusions?

Reviewer #1: Yes

Reviewer #2: Yes

2. Has the statistical analysis been performed appropriately and rigorously?

Reviewer #1: N/A

Reviewer #2: N/A

3. Have the authors made all data underlying the findings in their manuscript fully available?

Reviewer #1: Yes

Reviewer #2: Yes

4. Is the manuscript presented in an intelligible fashion and written in standard English?

Reviewer #1: Yes

Reviewer #2: Yes

5. Review Comments to the Author

Reviewer #1

R1#C#0: This manuscript presents a carefully conducted qualitative study exploring Catholic priests’ experiences and understandings of divine forgiveness (DF) within sacramental confession. The topic is original, theoretically engaged, and timely within the emerging psychological literature on divine forgiveness.

One of the strongest aspects of the manuscript is the rigorous and sustained engagement with the two central theoretical models (Fincham & May; Cook & Cowden). The authors not only present these models clearly and accurately in the Introduction, but also maintain them as analytic reference points throughout the Results and Discussion. The models are neither superficially cited nor merely decorative; rather, they structure the analytic logic of the study and are revisited critically in light of the data. This theoretical coherence significantly strengthens the manuscript and distinguishes it from purely descriptive qualitative work.

The study is methodologically transparent, the analytic process is clearly described, and the integration with existing theoretical models is thoughtful. The manuscript has clear potential for publication. However, several conceptual and interpretive issues require clarification before it can be considered suitable for publication in a general scientific journal such as PLOS ONE.

R1#C#0_Response: We thank the reviewer for their careful reading of the manuscript and for their thoughtful and constructive comments. We particularly appreciate the positive evaluation of the study’s originality, methodological transparency, and theoretical engagement, as well as the recognition of our sustained and rigorous use of the two central theoretical models throughout the manuscript.

We also greatly value the reviewer’s suggestions for improvement. We have carefully considered all the points raised and have revised the manuscript accordingly. Below, we respond to each comment in detail and indicate how the manuscript has been revised in response.

My comments focus on strengthening theoretical precision, interpretive restraint, and epistemological clarity.

R1#C#1: 1. Scope of Theoretical Claims

The manuscript proposes that the confessor may constitute a distinct “fourth vertex” in the relational spirituality model of personal sin. While this is a compelling interpretive insight, the current formulation risks overstating the implications of findings derived from a small and relatively homogeneous sample (N = 10 experienced male priests, predominantly from Spain).

I recommend moderating the strength of this claim. Rather than suggesting structural modification of the model, the argument would be more appropriately framed as a context-specific elaboration within sacramental Catholic settings, and a hypothesis-generating extension requiring further empirical examination. The contribution remains valuable, but the language should reflect its exploratory status.

R1#C#1_Response: We thank the reviewer for this helpful comment and agree that the original formulation may have overstated the theoretical implications of our findings.

In response, we have revised several passages in the Discussion to moderate the strength of our claims and more clearly reflect their exploratory and context-specific nature. Specifically, we now frame our interpretations as emerging from “this sample and context,” adopt more tentative language throughout (e.g., “appears to,” “may be experienced as”), and clarify that the role of the confessor is described from participants’ perspectives rather than as a generalizable structural feature. Importantly, the passage proposing the “fourth vertex” has been reformulated to emphasize that this is a tentative, hypothesis-generating extension, and we explicitly state that it should not be understood as a structural modification of the model but as a contextually grounded elaboration within sacramental Catholic settings (see revised Discussion section).

We have also aligned Fig 2 and its caption with this more cautious framing. The figure is now described as a contextually grounded, exploratory adaptation based on a small sample, and the confessor is presented as being provisionally conceptualized as a possible additional relational element. The description of the relational links has also been revised using more tentative language (see Fig 2’s caption). We hope these revisions ensure that the scope of our theoretical claims is appropriately aligned with the exploratory nature of the study.

R1#C#2: 2. Distinction Between Psychological Description and Theological Normativity

The manuscript generally maintains a psychological lens; however, in several passages—particularly in the Results and Discussion—the wording occasionally moves very close to doctrinal formulation rather than analytic description.

For example: In the Results, statements such as “You can only be certain of God’s forgiveness when the priest… says ‘I absolve you’” are presented with minimal analytic framing. While clearly derived from participant quotations, the surrounding text could more explicitly indicate that this reflects a participant belief structure rather than an ontological claim about the nature of forgiveness. In the Discussion, the description of the confessor as one who “sacramentally identifies with Christ and enacts DF” reads, at points, as a theological assertion rather than as an account of how priests conceptualize their role. A small shift in phrasing (e.g., “is understood by participants as…”) would maintain scientific neutrality. Similarly, when describing absolution as the moment in which the penitent “is reunited with God,” the manuscript could clarify that this is a phenomenological or doctrinally mediated interpretation rather than an empirically demonstrable event.

None of these instances undermine the study. However, because PLOS ONE is a general scientific journal, maintaining a consistently descriptive psychological register—especially when dealing with sacramental theology—would enhance epistemological clarity. The suggested revisions require only slight adjustments in phrasing, not substantive reanalysis.

R1#C#2_Response: We thank the reviewer for this thoughtful comment and agree that, in several passages, the wording could be read as moving too close to doctrinal formulation rather than remaining consistently within a descriptive psychological register.

In response, we have carefully revised the relevant passages in both the Results and Discussion to clarify that these statements refer to participants’ understandings, experiences, and belief structures, rather than to ontological claims about the nature of divine forgiveness. Specifically, we introduced more descriptive phrasing throughout (e.g., “was described by participants as,” “is understood within Catholic doctrine, and by participants, as,” “may be experienced as,” “appears in participants’ accounts as”) and adjusted several formulations that could have sounded overly theological or normative. We also revised instances referring to reconciliation with God and absolution to make clearer that these are phenomenological or doctrinally mediated interpretations rather than empirically demonstrable claims. Please see the revised manuscript with tracked changes/marked changes for the exact wording modifications in the Results and Discussion sections.

R1#C#3: 3. Cultural Framing and Representativeness

Although the study includes participants from four Spanish-speaking countries, six of the ten participants are based in Spain. At several points, the manuscript refers broadly to “Spanish-speaking contexts.” I recommend slightly narrowing this phrasing or explicitly ac

---

## [Editor Report · Decision Letter 1]

5 Apr 2026

Mediating Reconciliation With God: Exploring Divine Forgiveness Experiences During Confession Among Catholic Priests From Four Spanish-Speaking Countries

PONE-D-25-62289R1

Dear Dr. Rodríguez-González,

We’re pleased to inform you that your manuscript has been judged scientifically suitable for publication and will be formally accepted for publication once it meets all outstanding technical requirements.

Kind regards,

Rafael Galvão de Almeida, PhD.

Academic Editor

PLOS One

Additional Editor Comments (optional):

Please add the reviewers in the Acknowledgements section.
---

## [Editor Report · Acceptance letter]

PONE-D-25-62289R1

PLOS One

Dear Dr. Rodríguez-González,

I'm pleased to inform you that your manuscript has been deemed suitable for publication in PLOS One. Congratulations! Your manuscript is now being handed over to our production team.

Kind regards,

on behalf of

Dr. Rafael Galvão de Almeida

Academic Editor

PLOS One